# Diatom-Inspired Design: A New Ru-Based Photosystem for Efficient Oxygen Evolution

**DOI:** 10.3390/ma19010134

**Published:** 2025-12-30

**Authors:** Ambra Maria Cancelliere, Rosalia Maria Cigala, Mario Samperi, Catia Cannilla, Francesco Nastasi, Ileana Ielo, Giuseppina La Ganga, Giovanna De Luca

**Affiliations:** 1Department of Chemical, Biological, Pharmaceutical and Environmental Science, University of Messina, Via F. Stagno d’Alcontres 31, 98166 Messina, Italy; ambramaria.cancelliere@unime.it (A.M.C.); rosaliamaria.cigala@unime.it (R.M.C.); francesco.nastasi@unime.it (F.N.); giovanna.deluca@unime.it (G.D.L.); 2Interuniversitary Research Center for Artificial Photosynthesis (SOLARCHEM), University of Messina, Via F. Stagno d’Alcontres 31, 98166 Messina, Italy; 3National Research Council-Institute of Advanced for Energy Technology (CNR-ITAE), Via Comunale S. Lucia 5, 98126 Messina, Italy; mario.samperi@cnr.it (M.S.); catia.cannilla@cnr.it (C.C.); 4National Research Council-Institute for Microelectronics and Microsystems (CNR-IMM), Viale F. Stagno d’Alcontres 31, 98158 Messina, Italy

**Keywords:** photocatalysis, heterogeneous catalysis, composite material, ruthenium complex, diatomaceous earth, covalent grafting

## Abstract

The development of efficient and recyclable catalysts is a central pursuit in modern chemistry. Homogeneous catalysts, while effective, often suffer from challenges in separation and recovery, driving the exploration of heterogeneous systems. In this context, this study introduces a novel composite photocatalyst, Ru(bpy)_2_(bda)-Ru(bda)(cp)_2_@DE (PS/Cat@DE), synthesized by attaching a catalyst (Cat) and a photosensitizer (PS) to diatomaceous earth (DE). The hypothesis that covalently binding the photosensitizer and photocatalyst to the surface of DE could enhance their reactivity and may protect them from degradation was supported by the enhanced photocatalytic performance observed in this study. The composite materials and single components were characterized using UV-Vis and FTIR spectroscopy, as well as SEM, and EDS microscopy. Photocatalytic experiments demonstrated the significantly higher activity of the PS/Cat@DE material compared to equivalent concentrations of the single photosensitizer or photocatalyst components, indicating the crucial role of DE in promoting oxygen evolution.

## 1. Introduction

A central goal in modern chemistry is achieving environmentally sound chemical transformations, guided by the tenets of Green Chemistry [1,2]. Enhancing the efficiency of numerous industrial chemical reactions requires a deep understanding of both the reaction parameters and the characteristics of the catalytic agents themselves. This, in turn, drives the need to create and study catalysts that are not only highly active and selective but also capable of being recycled.

Most catalytic processes employ heterogeneous catalysis, where tiny particles of transition metals are spread across supports with large surface areas. These catalysts are essential to many industries, including those producing food, pharmaceuticals, and petrochemicals, and are responsible for roughly 90% of global chemical manufacturing [3,4]. Heterogeneous catalysis offers a cost-effective and environmentally sound approach. The catalytic activity takes place at specific sites on the catalyst, which is typically solid, non-organic material. Despite considerable advancements in creating supported catalysts on diverse materials like zeolites [5], clays [6], and silica [7], industrial applications often require catalysts that can withstand challenging reaction conditions [4].

Inspired by nature’s catalytic machinery, researchers have explored biomimetic catalysts, seeking to mimic the efficiency and selectivity of enzymes [8] and natural processes such as natural photosynthesis [9]. In natural photosynthesis, PSI [10] and PSII [11] play the role of collecting the light, transferring it in the reaction center as electronic energy. Mimicking the natural photosynthetic process is very hard due to the complexity of the natural systems. For this reason, to reach this goal, multicomponent devices are required [12,13]. In order to replicate the function of the natural systems, an artificial photosynthetic system requires a light harvesting unit (the so-called photosensitizer, PS) [14], a charge-separation component and two catalysts (Cat): one for the oxidation side (for O_2_ production, the bottle-neck of the water splitting process) and one for the reduction side (for H_2_ production, Figure 1) [15].

Considering only the oxidative side, in artificial photosynthesis, the photosensitizer absorbs light and switches to an excited state (Equation (1)), promoting oxidation by the sacrificial agent, the persulfate anion (Equation (2)). The sulfate radical formed oxidizes another photosensitizer molecule (Equation (3)). The oxidized species transfers an electron hole to the catalyst (Equation (4)). The sulfate radical regenerates the oxidized catalyst (Equation (5)). A total of two photons are required to accumulate a total of four electron holes within the catalyst (Equation (6)), which are required to oxidize one water molecule.PS + *hν* → *PS(1)*PS + S_2_O_8_^2−^ → PS^+^ + SO_4_^2−^ + SO_4_^−^(2)PS + SO_4_^−^ → PS^+^ + SO_4_^2−^(3)PS^+^ + Cat → PS + Cat^+^(4)Cat + SO_4_^−^ → Cat^+^ + SO_4_^2−^(5)Cat^4+^ + 2 H_2_O → Cat + O_2_ + 4 H^+^(6)

Ruthenium complexes, known for their versatile photophysical and redox properties, have emerged as promising candidates for both catalysts and photosensitizer subunits [14,16,17]. These complexes can be tailored for specific reactions through ligand design and offer a robust platform for driving photochemical processes. Their ability to absorb light and undergo electron transfer makes them ideal components in light-driven catalytic systems. However, like other homogeneous catalysts, challenges related to separation and recovery have limited their large-scale implementation.

To address these limitations, and to mimic the active site isolation found in natural systems, researchers have focused on immobilization of these ruthenium complexes on solid supports to enhance catalyst stability, facilitate separation, and can improve reaction selectivity. While supports such as synthetic silica [18], zeolites [19], MOFs [20], and graphene [21] have been explored, the selection of Diatomaceous earth (DE) is justified by its unique, naturally hierarchical porous structure. This bio-silica morphology facilitates the required spatial control and physical confinement more effectively than conventional synthetic silica, an approach confirmed by other studies using mesoporous materials to stabilize water oxidation catalysts [22,23], while simultaneously offering a cost-effective and abundant alternative [14,24,25,26]. Diatoms, common single-celled algae found in aquatic environments, contain porous cell walls, known as frustules, which are composed of silica and resemble amorphous silica [24,25,26]. When diatoms die, their frustules accumulate, forming DE, which is a readily available and inexpensive source of highly porous silica with potential for use in advanced materials [27,28,29,30]. The unique structure, transport properties, and biocompatibility of diatoms make them appealing for various uses, including catalysis, sensing, and filtration [27,31]. In the field of artificial photosynthesis, confining the photosensitizer and the catalyst in a limited space—for example, a silica nanocage—is a good strategy to increase the overall efficiency of the process and limit the decomposition of the components [32].

Importantly, the silanol groups on DE frustules can be readily modified chemically [33], allowing for the immobilization of various catalytic functionalities [34]. This prevents leaching of the catalytic species during reactions and provides control over both catalytic efficiency and selectivity.

To advance sustainable energy solutions, specifically artificial photosynthesis, overcoming the separation limitations of homogeneous ruthenium(II) complexes is mandatory. This study introduces a new composite photocatalyst, Ru(bpy)_2_(bda)-Ru(bda)(cp)_2_@DE, produced by attaching a catalyst (Ru(bda)(cp)_2_, Cat) and a photosensitizer (Ru(bpy)_2_(bda), PS) to the surface of DE. Considering the porosity of DE, and therefore the large surface area-to-volume ratio, we hypothesized that covalently binding PS and Cat to the surface could enhance their availability to react and, critically, potentially protect them from decomposition. The resulting composite was characterized using UV-Vis and FTIR spectroscopy, as well as SEM, and EDS microscopy. Photocatalytic experiments showed that the Ru(bpy)_2_(bda)-Ru(bda)(cp)_2_@DE material is significantly more active than the PS or Cat alone, without the DE support. This enhanced activity suggests that DE plays a critical role in promoting the evolution of oxygen.

## 2. Materials

Diatomaceous earth (DE) and distilled water (H_2_O_dist_) were purchased at VWR. DE used in this work exhibits a mixed morphology characterized by non-homogeneous micrometric dimensions. This natural siliceous material consists of a heterogeneous assembly of fossilized diatom frustules, displaying a variety of porous shapes and sizes that contribute to its inherently polydisperse structural nature. Ammonium hydroxide (NH_4_OH 30% *w*/*w*), ethanol (CH_3_CH_2_OH ≥ 99.5% reagent grade), and hydrogen peroxide (H_2_O_2_ 30% *w*/*w*) were purchased from Sigma Aldrich (Merk) (St. Louis, MO, USA). Sodium persulfate (Na_2_S_2_O_8_
≥ 98% reagent grade) used as sacrificial agent in photocatalysis was purchased from Riedel-de Haen. For the preparation of the phosphate buffer at pH 7, potassium phosphate dibasic (K_2_HPO_4_ ≥ 98% dried basis) was purchased from T.J. Baker and sodium hydroxide was purchased from Sigma Aldrich (Merk). cis-Bis(2,2′-bipyridine)dichlororuthenium(II) hydrate, 2,2′-bipyridine-4,4′-dicarboxylic acid, sodium bicarbonate (NaHCO_3_ ≥ 98% reagent grade), methanol (CH_3_OH ≥ 99.5% reagent grade), 6,6′-dimethyl-2,2′-bipyridine (≥99.5% reagent grade), sulfuric acid (H_2_SO_4_ ≥ 98% reagent grade), CrO_3_ (≥99.5% reagent grade), [Ru(DMSO)_4_Cl_2_] (≥99.5%), triethylamine ((CH_3_CH_2_)_3_N ≥ 99.5%), and 4-pyridine carboxylic acid (≥99.5% reagent grade), used for synthesis, were purchased from Sigma Aldrich (Merk) and used without any further purification. Anhydrous solvents were purchased from Sigma Aldrich (Merk) and used as received. The synthesis of the photosensitizer Ru(bpy)_2_(bda) in Figure 2a [35] and the catalyst Ru(bda)(cp)_2_ in Figure 2b [36] were performed by slightly modifying the procedure reported in the literature as follows.

## 3. Methods

### 3.1. Synthesis of Ru(bpy)_2_(bda) (PS)

cis-Bis(2,2′-bipyridine)dichlororuthenium(II) hydrate (200 mg, 0.41 mmol), 2,2′-bipyridine-4,4′-dicarboxylic acid (bda) (150 mg, 0.6 mmol), and sodium bicarbonate (0.2 g, 2.4 mmol) were dissolved in a mixture of methanol and water (*v*/*v* = 4:1). The solution was refluxed under an argon atmosphere for 24 h. After cooling, saturated aqueous ammonium hexafluorophosphate was added, resulting in a brown solid, which was then collected via filtration and air-dried. The solid was subsequently dissolved in acetonitrile, and diethyl ether was slowly diffused into the solution to promote crystallization. The final product was isolated by filtration and dried at room temperature [35].

### 3.2. Synthesis of 2,2′-Bipyridine-6,6′-dicarboxylate

A solution of 11 mL of concentrated H_2_SO_4_ and 6,6′-dimethyl-2,2′-bipyridine (1 g, 5.4 mmol) was prepared and CrO_3_ (3.23 g, 32 mmol) was added to this solution under stirring. The mixture was maintained at 70 °C for 1 h. The resulting dense green solution was then cooled in an ice bath. The precipitate formed was filtered, washed with water, and subsequently dried with diethyl ether to afford a white solid (1.25 g, yield 94%) [37].

### 3.3. Synthesis of Ru(bda)(cp)_2_ (Cat)

This synthesis was performed under N_2_ and anhydrous atmosphere. Ru(bda)(DMSO)_2_ (79 mg, 0.17 mmol) and 4-pyridine carboxylic acid (204 mg, 1.66 mmol) were dissolved in 10 mL of anhydrous methanol and heated to reflux. After twenty-four hours, the reaction mixture was allowed to cool to room temperature. The solution was then stored in a freezer overnight. The resulting brown precipitate was collected by filtration and washed with cold methanol. The product was obtained as a brown–orange solid (84 mg, yield 84%).

### 3.4. Functionalization of DE with Photocatalyst and Photosensitizer

DE was washed and activated using an alkaline Piranha solution. Then, 200 mg of DE was weighed and placed in a beaker with 20 mL of H_2_O_dist_. The mixture was heated to 80 °C under magnetic stirring. Once the temperature was reached, 10 mL of H_2_O_2_ and 10 mL of NH_4_OH were added. The solution was stirred for 20 min, then the DE was extensively washed with H_2_O_dist_ and dried in a ventilated oven at 50 °C overnight.

Then, 3 mg of Cat (Figure 2b) was weighed and dissolved in 10 mL of ethanol in a round-bottom flask under magnetic stirring. Subsequently, the solid DE was added to the solution, and the mixture was evaporated using a rotary evaporator. The DE powder collected after evaporation was washed three times with abundant ethanol and three times with abundant H_2_O_dist_. Following this, 18 mg of PS (Figure 2a) was weighed and dissolved in 10 mL of ethanol. The DE pre-functionalized with Cat was then added, and the mixture was evaporated using a rotary evaporator. The DE functionalized with both Cat and PS was subsequently washed three times with ethanol and three times with excess H_2_O_dist_. Figure 3 shows a schematic representation of the DE functionalization process.

### 3.5. UV-Vis Spectroscopy

All UV-Vis Absorption spectra were obtained with a Jasco V-560 spectrophotometer using water as solvent. Measurements were performed using 1 cm path length quartz cuvettes.

### 3.6. ATR-FTIR

Fourier Transform IR (FTIR) spectra were recorded in Attenuated Total Reflection (ATR) using a Thermo Scientific spectrophotometer model iS50 ATR (Parma, Italy). The scans were performed at a resolution of 4 cm^−1^, and the spectra were reported as percentage absorbance *vs*. wavelength (cm^−1^).

### 3.7. SEM-EDX

The surface morphology of the samples was investigated using an Ultra-High-Resolution Scanning Electron Microscope (UHR-SEM-FEG, Helios 5 UC DualBeam, Thermo Scientific) equipped with a Field Emission Gun (FEG), which provides high spatial resolution and improved image quality. SEM imaging was carried out on gold-coated samples using the in-lens SE/BSE detector, operating at an accelerating voltage of 10 kV and a beam current of 0.1 nA. Elemental analysis was performed by Energy Dispersive X-ray (EDX) spectroscopy.

### 3.8. Electrochemical Measurements

Electrochemical measurements were carried out with an Autolab multipurpose potentiostat with a GPES electrochemical interface in a three-electrode cell. The employed working electrode was an Amel glassy carbon with a diameter of 3 mm, the reference electrode was aqueous Ag/AgCl (3 M KCl aqueous solution), and the counter electrode was a platinum wire. The supporting electrolyte used was phosphate buffer (200 mM, pH 7.2) using a concentration of catalyst of 2 mM.

### 3.9. Photocatalytic Oxygen Evolution

Photocatalytic oxygen evolution was monitored using a Clark-type electrode system (Hansatech Oxygraph Plus, King’s Lynn, UK). This system employs an electrochemical sensor consisting of a platinum cathode and a silver anode, connected to a control unit that applies to a polarizing voltage. Oxygen present in the reaction solution above the cathode generates a current proportional to its concentration. This current is converted to a voltage, processed, and digitized before being displayed. The system was calibrated using oxygen-free (nitrogen-purged) and oxygen-saturated solutions. Air-saturated solution was used to fine-tune the calibration, targeting a reading of 19% oxygen. Oxygen concentration was recorded every 2.5 s using the OxyTrace+ software 1.0.48.

In a typical experiment a solution containing persulfate as the sacrificial agent (20 mM) in phosphate buffer (20 mM, pH 7.2) and the photoactive species were degassed under nitrogen flow and, after one hour and sensor calibration, the molecular photocatalyst at different concentrations was added. The mixture was irradiated with visible light (λ > 400 nm) using a xenon lamp with a cut off filter.

## 4. Results and Discussion

### 4.1. UV-Vis Spectroscopy

The photophysical characterization of PS and Cat in aqueous solution was performed.

Both species absorb in the visible range with characteristic transitions, but the PS absorbs more visible light, as expected, due to its higher molar extinction coefficient. The absorption spectra of Ru(II) polypyridine complexes (in black in Figure 4) are characterized by intense bands in the UV region due to ligand-centered (π→π*) transitions at 290 nm, and strong bands in the visible region attributed to spin-allowed MLCT transitions. Naturally, the energies at which MLCT transitions occur depend on the chemical environment around the metal center and the type of ligand coordination. Specifically, to determine which electronic transition corresponds to the lowest-energy ^1^MLCT state, one must identify the ligands that are more easily reduced and the metal(s) that are more easily oxidized. In the case of the mononuclear Ru species (PS), the only possible MLCT transitions are those from Ru to bpy (λ = 450 nm), represented by the black line in Figure 4.

Studies on the photophysical properties of ruthenium polypyridine complexes, well-documented in the literature since the 1980s [38,39,40,41], continue to hold significant scientific interest, with data further corroborated by computational studies [42].

### 4.2. ATR-FTIR

The FTIR spectrum of the PS (Figure 5) is dominated by vibrational modes originating from the Ru(II) center and the aromatic ligands (bpy and bda) containing carboxylic acid functions [43]. A broad, low-intensity band in the region of 3500–3000 cm^−1^ is attributed to the O-H stretching (*ν*(O-H)) vibrations of the carboxylic acid groups (-COOH) of the bda ligand. The sharp band centered at 1716 cm^−1^ is assigned to the two carbonyl stretching (*ν*(C=O)) of the -COOH group that is non coordinated. Complex features include multiple sharp bands in the 1600–1200 cm^−1^ region [44]. These are characteristics of the stretching vibrations of the C=C and C=N bonds within the pyridine aromatic rings of both the bpy and bda ligands. The shift in these bands compared to the free ligands confirms the coordination of the nitrogen atoms to the Ru(II) metal center. Bands in the 700–1100 cm^−1^ range are mainly associated with C-H out-of-plane bending and ring deformation modes of the substituted pyridine rings. The Metal–Ligand stretching modes Ru-N, which occur at 556, 445, and 423 cm^−1^, provide evidence of coordination [45].

The FTIR spectrum of Cat (Figure 6) exhibits key differences compared to the PS spectrum, reflecting its distinct ligand composition. The structure of Cat contains the bca ligand (bipyridine dicarboxylic acid) and two cp ligands (4-pyridine carboxylic acid). The most distinguishing features of the Cat spectrum are found in the C=O and carboxylate -COO^−^ stretching regions, reflecting a balance between protonated (-COOH) and coordinated (-COO^−^) groups. Specifically, a strong sharp band at 1715 cm^−1^ is attributed to the *ν*(C=O) stretch of the -COOH groups non coordinated, likely from the bca and cp ligands [46]. Strong bands throughout the 1640–1600 cm^−1^ range confirm the C=C and C=N stretching modes of the pyridine aromatic rings in the coordinated bca and cp ligands and the symmetric carboxylate stretching mode *ν*_s_(COO^−^) in the 1410–1360 cm^−1^ range. At 1266 cm^−1^, the mixed vibrations of the pyridine ring occur. Finally, the Ru-N stretching modes are present at 516, 471, and 429 cm^−1^, confirming the coordination of all *N*-heterocyclic ligands to the Ru(II) metal center, and the Ru-O stretching modes are present at 556 and 578 cm^−1^, confirming the coordination of the oxygen atoms of bca -COO^−^ groups [43,44,47].

### 4.3. SEM-EDX

The Scanning Electron Microscope (SEM) images (Figure 7) reveal the intricate morphology of the DE, with microscopic porous objects of different size and shape together with small particles of nanometric features.

To confirm the presence and loading of ruthenium, a key component of the Cat and PS, EDX analysis was carried out on functionalized DE (Figure 8). The accompanying EDX spectrum highlights a small peak corresponding to the ruthenium L line at 2.56 keV, alongside the other peaks related to the different elements present in the DE matrix. The quantification of the amount of ruthenium, reported as weight percentage composition as shown in the inset table in Figure 8, provides definitive evidence of functionalization, highlighting the successful integration of Cat and PS units on the DE surface.

### 4.4. Electrochemical Measurements

Electrochemical characterization of the two species in solution, performed via cyclic voltammetry and Differential Pulse Voltammetry, confirmed their respective oxidation potentials. The data reveal all oxidation processes associated with the catalyst (+0.47, +0.69, +0.82, and +1.09 V vs. Ag/AgCl) showed in Figure 9, while the photosensitizer (PS) exhibits a single reversible oxidation wave at a more positive potential (+1.19 V vs. Ag/AgCl) showed in Figure 10.

This potential difference is of fundamental thermodynamic significance, as the oxidation potential of the PS is more positive than that of the catalyst, indicating that hole scavenging between the two species is thermodynamically favored (see Equation (4)).

Additional electrochemical measurements, including cyclic voltammetry in buffer solution, validated the catalytic activity of the catalyst, demonstrating that water oxidation in the presence of the catalyst occurs at a lower overpotential compared to the oxidation in pure solvent (Figure 11). This confirms that the catalyst effectively facilitates oxygen evolution, enhancing the overall efficiency of the system under the tested conditions.

All electrochemical measurements are reported in Table 1 and compared with literature data for model compounds.

### 4.5. Photocatalytic Oxygen Evolution

Considering the increasing photocatalytic performances for confined system [32] or to evaluate the possibility of depositing this protected material on the surface of a photoanode [48], we decided to test the photocatalytic performance of PS/Cat@DE.

The photocatalytic performance of the PS/Cat@DE system was evaluated by monitoring, with a Clark-type electrode, the amount of dissolved oxygen in 2 mL of solution. Figure 12 shows the oxygen production by using the integrated system PS/Cat@DE in comparison to a mixture of the separated components free in solution using persulfate as a sacrificial electron acceptor. To compare the photocatalytic performance of the heterogeneous system (PS/Cat@DE) and to evaluate the effect of the space confinement, we calculate the concentration of photosensitizer and catalyst by assuming they are uniformly distributed inside the DE. This assumption was necessary to ensure equivalent light-harvesting abilities and catalyst amounts for a meaningful comparison between the integrated and homogeneous systems. Although direct spatial mapping of the ruthenium complexes across the DE structure was not performed, the success of the covalent grafting procedure and the subsequent extensive washing steps provide strong support for the robust immobilization of PS and Cat onto the DE surface. Confirmation of the overall loading was verified through EDX analysis, which detected the presence of ruthenium on the functionalized DE. The enhanced activity (eight-fold greater turnover number) observed for the PS/Cat@DE system strongly suggests that the effective concentration of active sites available for reaction is significantly higher in the integrated material, overcoming the diffusion limitations present in the homogeneous mixture. We acknowledge that potential non-uniformity in distribution could impact the exact localized concentration values, representing a limitation in the calculated turnover number (TON), which reflects the performance under the most favorable conditions for confinement. As it is possible to observe in Figure 12, the activity of the system PS/Cat@DE is eight times bigger than the corresponding homogeneous mixed system. To test the catalyst’s activity, we also calculate the TON that reached the value of 30 in the case of PS/Cat@DE. As in the kinetics of oxygen production, the TON is eight times bigger; in fact, the TON of the homogeneous mixed system is 4, as well as the initial rate of oxygen production. The better photocatalytic performance could be explained by considering that the space proximity of the photosensitizer and catalyst units in the diatomaceous earth can overcome the limits imposed by diffusion instead of being present in the homogeneous mixed system.

To accurately simulate the matrix effect of the heterogeneous system and to compare it with the homogeneous system, we conducted experiments designed to replicate the light-scattering effects caused by the diatomaceous earth (DE). These experiments were aimed at ensuring that the observed photocatalytic performance was genuinely attributable to the functionalized system comprising the photosensitizer (PS) and catalyst (Cat), rather than being influenced by light scattering from the DE matrix.

Our findings indicate that the photocatalytic activity of the heterogeneous system demonstrated superior efficiency when functionalized with the PS and Cat. Importantly, the negligible amount of oxygen produced when utilizing the DE system alone supports the conclusion that the light-scattering effect does not contribute to enhancing the photocatalytic performance. This evidence suggests that the matrix provided by the DE does not obscure the data, confirming that the efficiency gains observed are a direct result of the photocatalytic components rather than an artifact of scattering effects.

Also, the Cat@DE system was tested under the same experimental conditions using an external PS. In this case oxygen was not detected because the PS, without the appropriate activation and functionalization process, shows some difficulty going inside the diatomaceous pores if it is homogeneously dispersed in solution. This highlights that simply combining the PS, Cat, and diatomaceous earth is insufficient for effective assembly and function; the specific functionalization process we employed is absolutely critical for achieving the desired photocatalytic activity. Given the substantially increased performance observed only in the PS/Cat@DE system, we hypothesize that this enhanced activity results from the spatial confinement, which may also contribute to protecting the catalytic components from decomposition. Further studies, including leaching and cycling tests, are required to confirm this proposed protective mechanism.

## 5. Conclusions

In the broader context of developing efficient and sustainable catalytic systems, this study successfully produced a new composite photocatalyst, PS/Cat@DE, by attaching a catalyst and a photosensitizer to the surface of diatomaceous earth. The motivation behind this approach stems from the challenges associated with homogeneous catalysts, particularly concerning separation and recovery, which has spurred research into immobilizing catalysts on solid support. Diatomaceous earth, with its unique structural and chemical properties, presents a compelling alternative as a support material. The hypothesis that covalently binding the photosensitizer and photocatalyst to the surface of DE could enhance their reactivity and potentially mitigate degradation was supported by the resulting eight-fold increase in the TON compared to the homogeneous system. The characterization of the photocatalyst and photosensitizer was performed in solution using UV-Vis spectroscopy, and in solid state using ATR-FTIR to verify the effective success of the synthesis and stability of the Cat and PS. Furthermore, SEM-EDX analyses were conducted to observe the morphology of the DE frustules and the effective functionalization of the DE with Cat and PS. Photocatalytic experiments demonstrated significantly higher activity of the PS/Cat@DE material compared to the photosensitizer or photocatalyst alone, indicating the crucial role of DE in promoting oxygen evolution.

## Figures and Tables

**Figure 1 materials-19-00134-f001:**
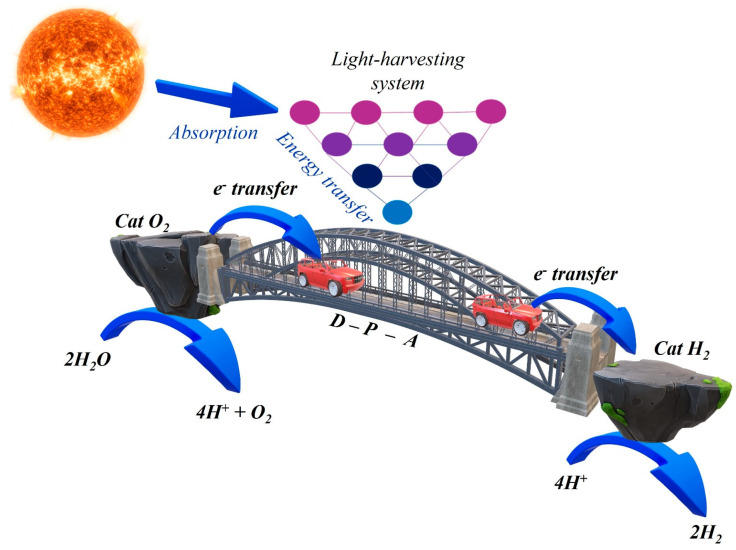
Three-dimensional stylized cartoon representation of an artificial photosynthetic multicomponent device, illustrating the light-harvesting system coupled with catalysts for the production of oxygen and hydrogen from water.

**Figure 2 materials-19-00134-f002:**
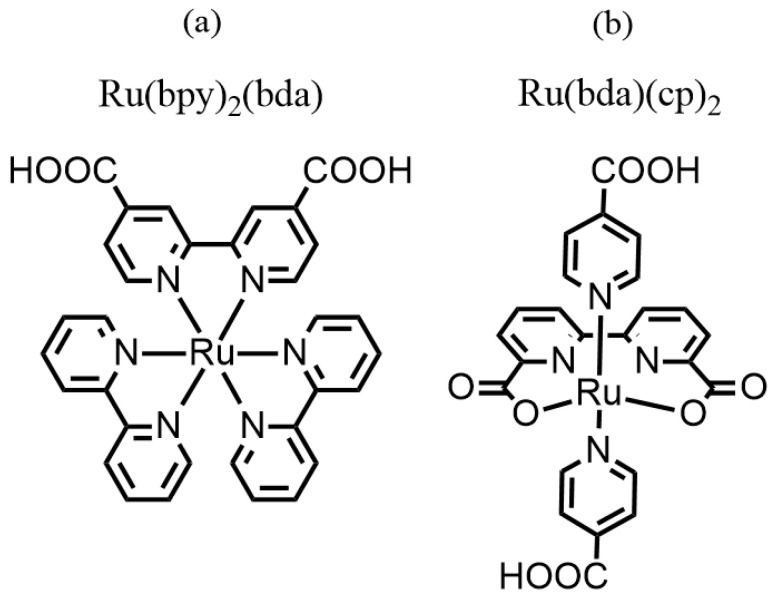
Chemical structure of (**a**) photosensitizer (PS) Ru(bpy)_2_(bda) and (**b**) photocatalyst (Cat) Ru(bda)(cp)_2_, both featuring carboxylic acid groups for surface anchoring.

**Figure 3 materials-19-00134-f003:**
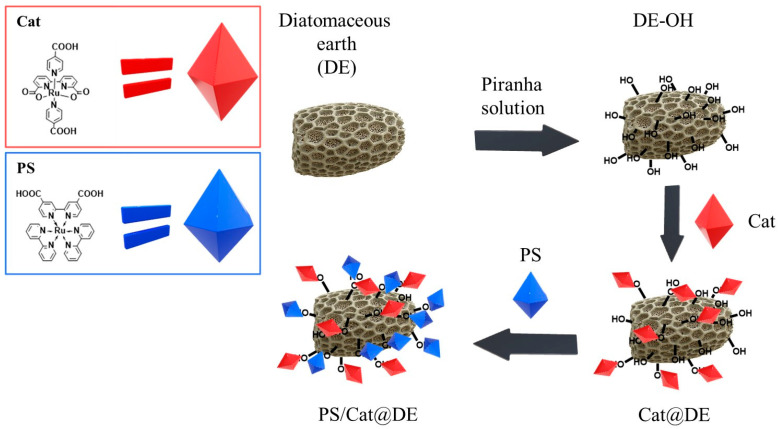
Stepwise schematic representation of PS/Cat@DE synthetic workflow, including DE surface activation to maximize silanol groups (DE-OH), followed by the sequential covalent grafting of the catalyst and photosensitizer.

**Figure 4 materials-19-00134-f004:**
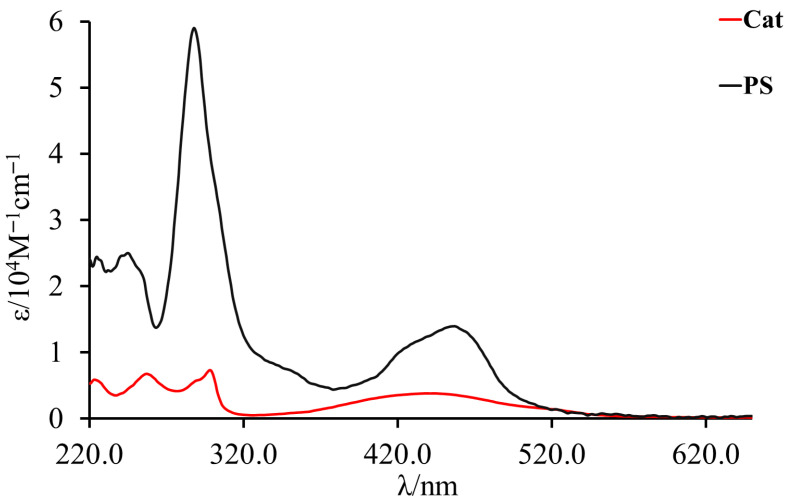
UV-Vis spectra of PS (black line) and Cat (red line) in 20 mM phosphate buffer (pH 7.2). The spectra highlight the intense MLCT transitions in the visible region (~420–450 nm) and ligand-centered transitions in the UV region (~290 nm).

**Figure 5 materials-19-00134-f005:**
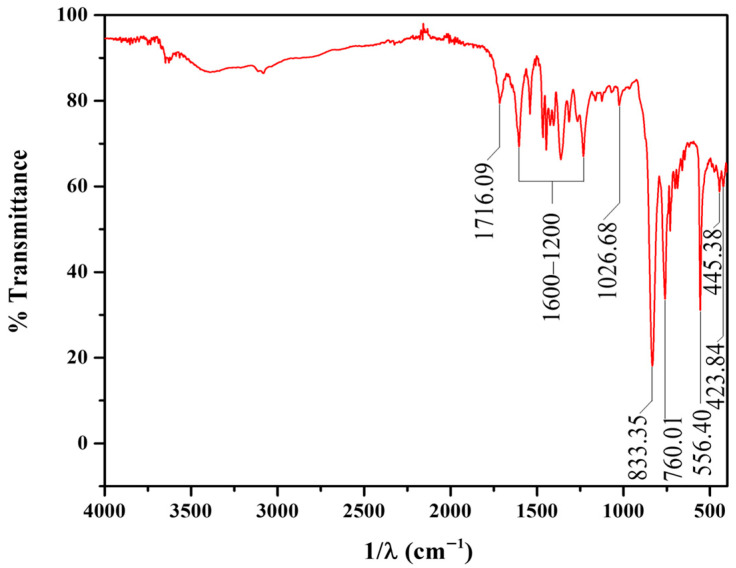
FT-IR spectrum of PS.

**Figure 6 materials-19-00134-f006:**
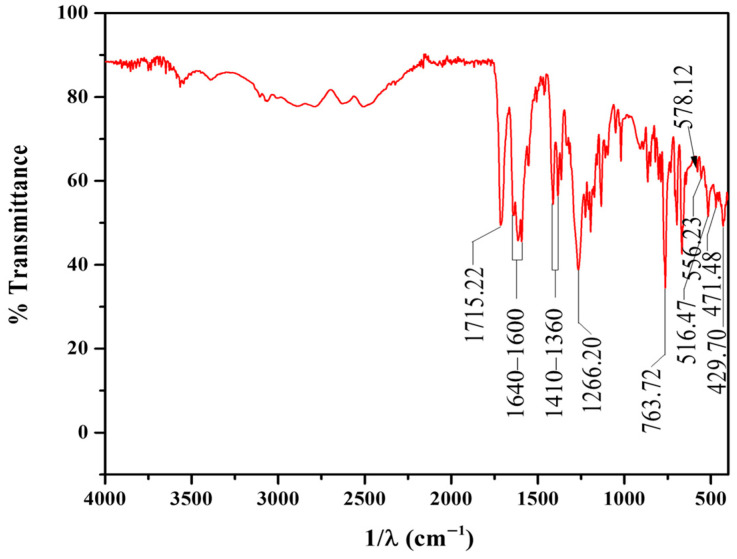
FT-IR Spectrum of Cat.

**Figure 7 materials-19-00134-f007:**
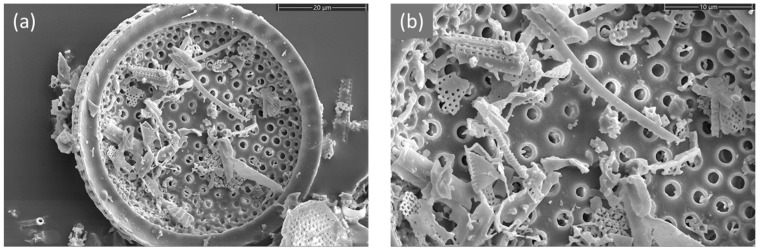
SEM micrographs of the raw DE structure at (**a**) magnification of 5kx and scale of 20 μm showing fossilized frustule morphology; (**b**) magnification of 10kx and scale of 10 μm detailing the hierarchical porosity and nanometric features.

**Figure 8 materials-19-00134-f008:**
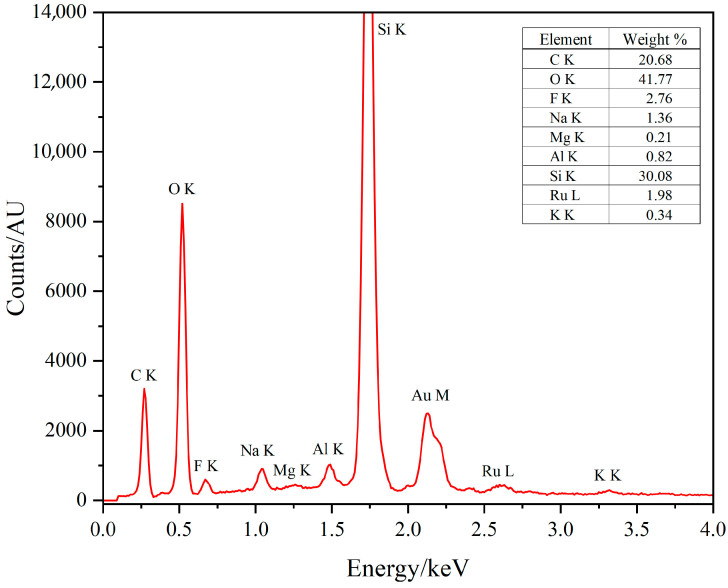
EDX spectrum of PS/Cat@DE, with table in inset reporting the weight percentage composition (the Au peak was excluded from quantification).

**Figure 9 materials-19-00134-f009:**
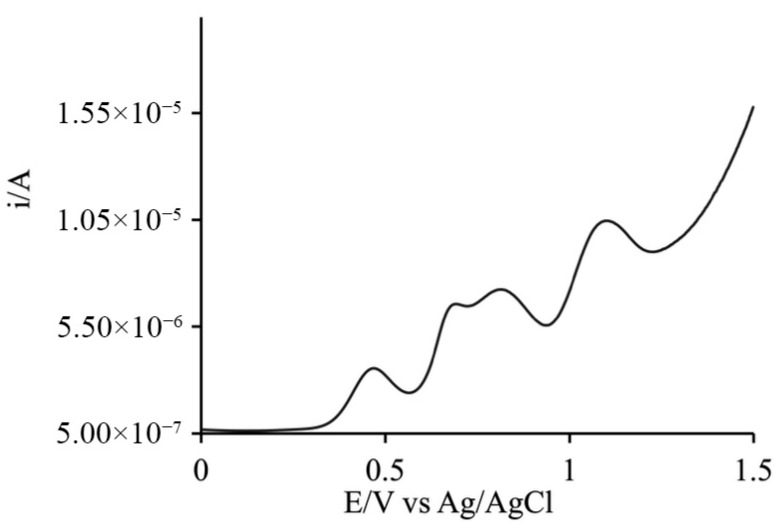
DPV experiment of Cat (2 mM) in 200 mM phosphate buffer, resolving the four sequential oxidation processes at +0.47, +0.69, +0.82, and +1.09 V vs. Ag/AgCl.

**Figure 10 materials-19-00134-f010:**
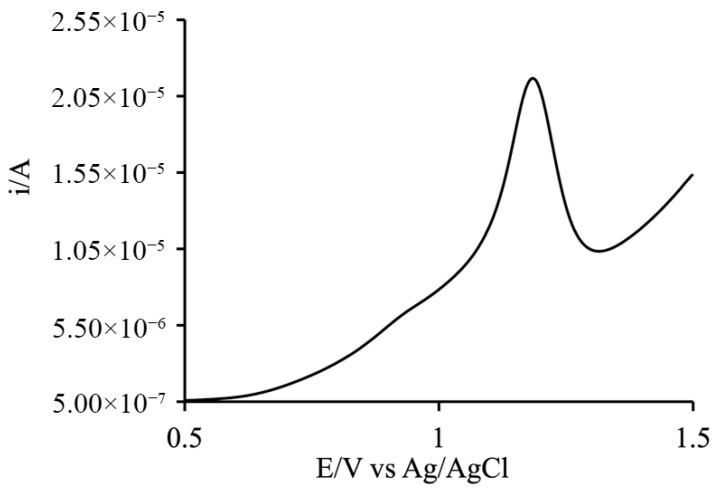
DPV of PS in 200 mM phosphate buffer, displaying a single reversible oxidation wave at +1.19 V vs. Ag/AgCl. This potential ensures that hole scavenging from the Cat is thermodynamically favored.

**Figure 11 materials-19-00134-f011:**
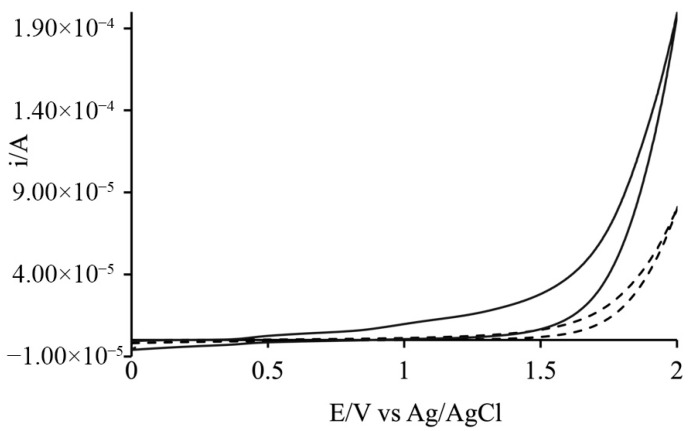
CV comparison in 200 mM phosphate buffer showing that the presence of Cat (solid line) significantly lowers the overpotential for water oxidation compared to the catalyst-free buffer (dashed line).

**Figure 12 materials-19-00134-f012:**
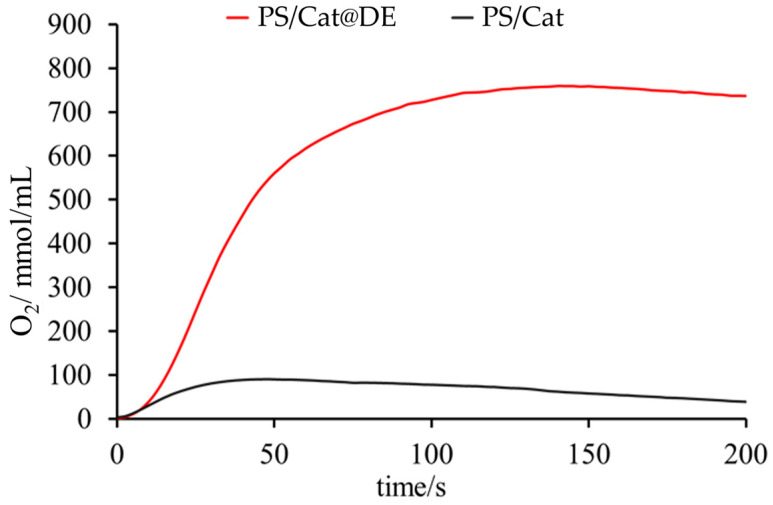
Comparative kinetics of photocatalytic oxygen evolution vs. time for PS/Cat@DE (2.5 × 10^−5^ M of Cat e 1.3 × 10^−4^ M of PS, red line) and the separated system (1 × 10^−4^ M of Ru(bpy)_3_ and 2.5 × 10^−5^ M of Cat, black line) in phosphate buffer (20 mM, pH = 7.2) in presence of Na_2_S_2_O_8_ (20 mM) as sacrificial electron acceptor. Irradiation lamp, λ > 400 nm. Detection of oxygen measurement: Clark electrode.

**Table 1 materials-19-00134-t001:** Oxidation potentials in buffer solution at room temperature. Potential is reported vs. Ag/AgCl.

**Compound**	**E_ox1_/V**	**E_ox2_/V**	**E_ox3_/V**	**E_ox4_/V**
[Ru(bpy)_3_]^2+^				+1.10
Ru(bda)(bpy)_2_				+1.19
Ru(bda)(pic)_2_ [16]	+0.47	+0.71	-	+1.05
Ru(bda)(cp)_2_	+0.47	+0.69	+0.82	+1.09

## Data Availability

The original contributions presented in this study are included in the article. Further inquiries can be directed to the corresponding authors.

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
