# Peer review of "Materials2026, 19(1), 134;https://doi.org/10.3390/ma19010134"

_materials, 2025, doi:10.3390/ma19010134_

Round 1

Reviewer 1 Report

Comments and Suggestions for Authors

Dear Authors,

   Please find the attached comments.

Thank you

Author Response

The research work by Ambra Maria Cancelliere et.al. reports on Diatom-inspired design: a new Ru-based photosystem for efficient oxygen evolution.

The manuscript represents the Diatom-inspired design: a new Ru-based photosystem for efficient oxygen evolution. The manuscript in general need to be modified and rewritten. However, some issue needs to be solved before fully accepting it. I recommend major revision according to the following comments,

Comments 1: In the equation 1 to 6 please clarify the oxidation states clearly. Moreover equation 6 to be corrected and justified. Specificare nella descrizione le eq sono generiche

Response 1: Thank you for your comments and for the opportunity to clarify these chemical processes. It is important to note that the equations provided in this section are intended to be generic representations of the electron transfer mechanisms. They describe the fundamental behavior of an arbitrary photosensitizer (PS) and a general catalyst (Cat), establishing the theoretical framework for photocatalysis before we delve into the specific ruthenium(II) complexes used in our experimental work later in the introduction. Furthermore, we have corrected a typographical error in equation 4, where the species was previously mislabeled as C+. This has been updated to Cat+ to maintain consistency with the nomenclature used for the catalyst throughout the text.

Comments 2: Introduction is not precisely written, there is a gap between the objective of the work.

Response 2: Thank you for your valuable feedback regarding the flow of the introduction. We appreciate the observation that the transition toward the specific objectives of the work could be more seamless. To address this gap and better define the purpose of our study, we have revised the text to include the following statement: "To advance sustainable energy solutions, specifically artificial photosynthesis, overcoming the separation limitations of homogeneous ruthenium(II) complexes is mandatory." This addition serves to bridge the theoretical background of photocatalytic cycles with the practical necessity of our research, highlighting why our specific approach to catalyst immobilization or recovery is a critical step forward for the field.”

Comments 3: Throughout the manuscript language has to be improved. More grammatical errors. For example, section 4.1, “there was performed”

Response 3: Thank you for your constructive feedback regarding the language and grammar throughout the manuscript. We appreciate your attention to detail, as it helps improve the overall clarity of our work. Please be informed that the English in the manuscript has been thoroughly revised using a premium AI editing service to ensure professional standards. Specifically, we have addressed the grammatical errors you highlighted by implementing several corrections. In Section 4.1, the phrase "There was performed the photophysical characterization of PS and Cat in aqueous solution" has been corrected to "The photophysical characterization of PS and Cat in aqueous solution was performed" to ensure proper sentence structure. In Section 4.5, we have updated the text to: "Considering the increasing photocatalytic performance for confined systems or aiming to evaluate the possibility of depositing this protected material on the surface of a photoanode, we decided to test the photocatalytic performance of PS/Cat@DE." Furthermore, in the introduction at lines 60-61, we corrected the original phrasing "...an artificial photosynthetic system requires a light harvesting units..." to the singular form "a light harvesting unit" to maintain correct subject-verb agreement.

Comments 4: Sub heading section 3.6 to 3.8 is not necessary as a separate paragraph. It can be included in the result and discussion part.

Response 4: Thank you for your suggestion regarding the organization of the manuscript. We appreciate your perspective on streamlining the text to improve the flow of the results and discussion. Regarding the placement of subsections 3.6 to 3.8, we initially followed the specific template provided by the journal Materials, which encourages a detailed description of the methods used. However, if you feel that these sections are superfluous to the main body of the paper or disrupt the narrative, we would be happy to relocate this technical information to the Supporting Materials. This would allow us to maintain the methodological rigor required by the journal while keeping the main text focused on the primary results and discussion.

Comments 5: Ligand centered (π→π*) transitions are absorbed at which wavelength? The description is not clear about Ru (II) MLCT transitions. Compare the results with other spectroscopic techniques. Which electronic transition is responsible for intense UV bands observed for Ru (II) complexes.

Response 5: Thank you for your valuable feedback regarding our manuscript. To clarify, the ligand-centered (π→π*) transitions are typical transition involving polipiridine ruthenium (II) complexes in the UV region and are typically more intense than the MLCT (Metal to Ligand Charge Transfer) transitions, which occur in the visible part of the spectrum. The latter are considered spin-forbidden transitions, resulting in lower intensity. Additionally, if we want a comparison with other spectroscopic techniques, it can be effectively made in relation to the MLCT transition from the metal to the ligand. This transition is characteristic of the oxidation state (+2) of the metal and is commonly observed in the DPV experiment of the photosensitizer (PS), as depicted in Figure 10, where the oxidation process of the metal is evident.

Comments 6: Is there any theoretical evidence to compare the results obtained from UV and FTIR? The authors could interpret metal to ligand charge transfer in Ru (II) polypyridyl complexes.

Response 6: Thank you for your insightful comment. There is a substantial body of literature, dating back to the 1980s, which thoroughly explains the photophysical properties of Ru(II) polypyridyl complexes, including detailed discussions on metal-to-ligand charge transfer (MLCT) mechanisms. To enhance clarity and provide comprehensive information, we will incorporate the following references that elucidate these properties further. (Meyer, T. J. Acc. Chem. Res.1989, 22, 163.; Sun, L.; Hammarström, L.; Åkermark, B.; Styring, S. Chem. Soc. Rev. 2001, 30, 36.; Barigelletti, F.; Flamigni, L. Chem. Soc. Rev.2000, 29, 1.; Juris, A.; Balzani, V.; Barigelletti, F.; Campagna, S.; Belser, P.; von Zelewsky, A. Coord. Chem. Rev.1988, 84, 85.; Sauvage, J.-P.; Collin, J.-P.; Chambron, J.-C.; Guillerez, S.; Coudret, C.; Balzani, V.; Barigelletti, F.; De Cola, L.; Flamigni, L. Chem. Rev.1994, 94, 993.) In particular, information is provided on the computational calculations carried out in the following work of literature (https://doi.org/10.1016/j.chemphys.2008.05.010), which offers the following insights: The infrared (IR) and Raman spectra of [M(bpy)₃]²⁺ complexes (where M = Fe and Ru) were generated based on the optimized geometries, utilizing a scaled quantum chemical force field. These results were then compared to a previous normal coordinate analysis of [Ru(bpy)₃]²⁺, which relied solely on experimental data and a simplified model. The computational results align well with the experimental findings, and the potential energy distributions derived from the normal coordinate analyses provide a comprehensive understanding of the vibrational spectra for both complexes.

Comments 7: What about the particle size? Line no 342 to 347 is not clear to understand.

Response 7: We thank you for your suggestion and have replaced the unclear paragraph with the following: “To accurately simulate the matrix effect of the heterogeneous system and to compare it with the homogeneous system, we conducted experiments designed to replicate the light-scattering effects caused by the diatomaceous earth (DE). These experiments were aimed at ensuring that the observed photocatalytic performance was genuinely attributable to the functionalized system comprised of the photosensitizer (PS) and catalyst (Cat), rather than being influenced by light scattering from the DE matrix. Our findings indicate that the photocatalytic activity of the heterogeneous system demonstrated superior efficiency when functionalized with the PS and Cat. Importantly, the negligible amount of oxygen produced when utilizing the DE system alone supports the conclusion that the light-scattering effect does not contribute to enhancing the photocatalytic performance. This evidence suggests that the matrix provided by the DE does not obscure the data, confirming that the efficiency gains observed are a direct result of the photocatalytic components rather than an artifact of scattering effects.” Further, In “materials” paragraph we add the following sentence: “DE used in this work exhibits a mixed morphology characterized by non-homogeneous micrometric dimensions. This natural siliceous material consists of a heterogeneous as-sembly of fossilized diatom frustules, displaying a variety of porous shapes and sizes that contribute to its inherently polydisperse structural nature.”

Comments 8: Does the SEM image provide DE morphology after functionalization? More explanation about the results should be addressed for all the techniques used in the article.

Response 8: Thank you for your insightful question regarding the interpretation of the SEM results and the necessity for a more detailed discussion of our characterization techniques. The SEM images were primarily utilized to reveal the intrinsic morphology of the diatomaceous earth (DE). As shown in the micrographs, the material displays the intricate, porous structure characteristic of DE frustules, which consist of microscopic porous objects of varying sizes and shapes, alongside smaller particles exhibiting nanometric features. This aligns with the non-homogeneous micrometric description provided in the Materials section. It is important to clarify that the SEM images alone do not provide direct visual evidence of the successful covalent attachment of the ruthenium complexes. Instead, EDX spectroscopy was performed on the functionalized composite (PS/Cat@DE) to confirm the loading of the catalyst and photosensitizer onto the support. The EDX spectrum provides definitive evidence of functionalization by highlighting a small peak corresponding to the Ruthenium L line at 2.56 keV. The quantification of Ruthenium, reported as a weight percentage, confirms the successful integration of the Cat and PS units onto the DE surface. Furthermore, we have expanded the discussion on the photocatalytic results. The enhanced Photocatalytic Oxygen Evolution activity is attributed to the spatial proximity of the PS and Cat units within the DE framework. This strategic arrangement allows the system to overcome the diffusion-limited kinetics that typically hinder the performance of homogeneous mixtures. Control experiments demonstrated that a simple physical mixture of PS, Cat, and DE is insufficient to achieve these results, highlighting that the specific covalent functionalization process is absolutely critical for the observed catalytic performance.

Reviewer 2 Report

Comments and Suggestions for Authors

In this paper, the authors present a new composite photocatalytic material based on diatomaceous earth that has been functionalised with two ruthenium complexes: a photosensitiser and a catalytic component. The article demonstrates that the covalent fixation of both complexes on the silica frustule's surface not only organises their arrangement, but also significantly enhances the catalytic oxygen evolution efficiency compared to a homogeneous system. 
While the work is of interest to various fields of science, there are some comments:
1. The authors highlight the unique structure of diatomaceous earth, yet fail to provide a direct comparison with other silica carriers, making it challenging to evaluate the merits of the proposed approach. Including such information in the introduction would significantly improve the work.
2. The materials should indicate the manufacturer of the components uniformly.
3. In line 173, "Error! Reference source not found" needs to be corrected.
4. The claim about preventing catalyst degradation is not confirmed by experimentation — there are no cyclic tests, Ru leaching analysis or restart tests. Either experimental data should be added or this claim should be removed.
5. The authors assume a uniform distribution of PS and Cat within DE, which has not been proven. Direct measurement of concentrations is required, or at least a discussion of the errors of such an assumption.
6. Photodegradation is an important issue for ruthenium complexes. A brief discussion of stability under experimental conditions should therefore be included.

Author Response

In this paper, the authors present a new composite photocatalytic material based on diatomaceous earth that has been functionalised with two ruthenium complexes: a photosensitiser and a catalytic component. The article demonstrates that the covalent fixation of both complexes on the silica frustule's surface not only organises their arrangement, but also significantly enhances the catalytic oxygen evolution efficiency compared to a homogeneous system.

While the work is of interest to various fields of science, there are some comments:
Comments 1: The authors highlight the unique structure of diatomaceous earth, yet fail to provide a direct comparison with other silica carriers, making it challenging to evaluate the merits of the proposed approach. Including such information in the introduction would significantly improve the work.

Response 1: Thank you for this constructive suggestion. We agree that providing a comparative context for the choice of support strengthens the rationale behind our methodology. To address this, we have updated the introduction to include the following sentence, supported by relevant literature: "While supports such as synthetic silica [18], zeolites [19], MOFs [20], and graphene [21] have been explored, the selection of Diatomaceous Earth (DE) is justified by its unique, naturally hierarchical porous structure. This bio-silica morphology facilitates the required spatial control and physical confinement more effectively than conventional synthetic silica, an approach confirmed by other studies using mesoporous materials to stabilize water oxidation catalysts [22,23], while simultaneously offering a cost-effective and abundant alternative [24–27]."

Comments 2: The materials should indicate the manufacturer of the components uniformly.

Response 2: Thank you for your valuable feedback regarding the consistency of our documentation. We agree that a uniform presentation of chemical sources is essential for the reproducibility of the study. We have updated the Materials section to ensure that the manufacturer and origin of all components and reagents are indicated consistently throughout the text. Each chemical, solvent, and support material now includes the manufacturer's name and location in a standardized format.

Comments 3: In line 173, "Error! Reference source not found" needs to be corrected.

Response 3: Thank you for bringing this to our attention. Please be informed that the bibliography was managed using Zotero, and it appears there was a technical formatting issue during the final export process. We have now addressed this inconsistency and carefully corrected the citations and reference list to ensure they align perfectly with the journal's required style.

Comments 4: The claim about preventing catalyst degradation is not confirmed by experimentation — there are no cyclic tests, Ru leaching analysis or restart tests. Either experimental data should be added or this claim should be removed.

Response 4: Thank you for this critical observation regarding the stability claims of our system. We acknowledge that the initial phrasing may have been interpreted as a definitive conclusion rather than a mechanistic hypothesis based on the observed catalytic performance. As you correctly pointed out, while the PS/Cat@DE system exhibited a significantly higher activity, yielding an 8-fold increase in the TON compared to the homogeneous mixture, our current study does not include explicit experimental data such as ICP-MS leaching analysis, cyclic reuse tests, or restart experiments to conclusively prove the prevention of degradation. The enhanced performance suggests that the DE framework provides a protective environment and overcomes diffusion limits, but we agree that direct evidence of long-term stability is required for such a claim. In response to your feedback, we have revised the manuscript throughout to clarify that this is a hypothesis supported by indirect evidence, rather than a confirmed experimental fact. The following changes have been implemented:

Abstract: We have updated the text to state that the covalent binding "may protect them from degradation," and that the hypothesis was supported by the "enhanced photocatalytic performance observed in this study."

Introduction: The language has been shifted to emphasize that we "hypothesized" that covalent binding could "potentially protect them from decomposition."

Discussion (Section 4.5): We have added a clarifying sentence: "Further studies, including leaching and cycling tests, are required to confirm this proposed protective mechanism." This acknowledges the current experimental boundaries while still discussing the implications of the 8-fold TON increase.

Conclusions: We have refined this section to state that the hypothesis regarding mitigated degradation was supported by the "resulting 8-fold increase in the TON compared to the homogeneous system," clearly linking the conclusion to the specific data collected.

Comments 5: The authors assume a uniform distribution of PS and Cat within DE, which has not been proven. Direct measurement of concentrations is required, or at least a discussion of the errors of such an assumption.

Response 5: Thank you for this insightful critique. We agree that the assumption of a perfectly uniform distribution of PS and Cat within the DE structure is a significant point that warrants further clarification, particularly as it forms the basis for our TON calculations. While our protocol included extensive washing steps to remove any non-covalently bound species and EDX analysis confirmed the presence of ruthenium across the bulk material, we acknowledge that we did not perform direct spatial mapping or analysis of the washing waters. To address this and improve the transparency of our methodology, we have revised Section 4.5 to explicitly discuss this assumption and its potential impact on the data. The following text has been inserted into the Photocatalytic Oxygen Evolution section: "To compare the photocatalytic performance of the heterogeneous system (PS/Cat@DE) and to evaluate the effect of space confinement, we calculated the concentration of the photosensitizer and catalyst by assuming they are uniformly distributed inside the DE. This assumption was necessary to ensure equivalent light-harvesting abilities and catalyst amounts for a meaningful comparison between the integrated and homogeneous systems. Although direct spatial mapping of the ruthenium complexes across the DE structure was not performed, the success of the covalent grafting procedure and the subsequent extensive washing steps provide strong support for the robust immobilization of PS and Cat onto the DE surface. Confirmation of the overall loading was verified through EDX analysis, which detected the presence of ruthenium on the functionalized DE. The enhanced activity (8-fold greater TON) observed for the PS/Cat@DE system strongly suggests that the effective concentration of active sites available for reaction is significantly higher in the integrated material, overcoming the diffusion limitations present in the homogeneous mixture. We acknowledge that potential non-uniformity in distribution could impact the exact localized concentration values, representing a limitation in the calculated TON, which reflects the performance under the most favorable conditions for confinement." This revision ensures that the reader is aware of the limitations of the uniform distribution model while emphasizing the strong experimental evidence, namely the 8-fold TON increase, that supports the effectiveness of our immobilization strategy.

Comments 6: Photodegradation is an important issue for ruthenium complexes. A brief discussion of stability under experimental conditions should therefore be included.

Response 6: Thank you for this critical observation regarding the stability of ruthenium complexes. We fully agree that photodegradation is a key challenge in this field and that a transparent discussion of stability under experimental conditions is essential.

We acknowledge that our initial phrasing may have been interpreted as a definitive conclusion rather than a mechanistic hypothesis based on the observed catalytic performance. While the PS/Cat@DE system exhibited a significantly higher activity—yielding an 8-fold increase in the TON compared to the homogeneous mixture—we recognize that this study does not currently include explicit experimental data, such as ICP-MS leaching analysis, cyclic reuse tests, or restart experiments, to conclusively prove the prevention of degradation. The enhanced performance suggests that the DE framework provides a protective environment and overcomes diffusion limits, but direct evidence of long-term stability is necessary for such a definitive claim.

In response to your feedback, we have revised the manuscript to clarify that this is a hypothesis supported by indirect evidence.

Reviewer 3 Report

Comments and Suggestions for Authors

The novelty of this work is an improvement of photocatalytic water oxidation by attaching of the known water oxidation catalyst Ru(bda)(cp)2 and the photosensitizer Ru(bpy)2(bda) to the surface of diatomaceous earth (DE).  The common protocol of photoinduced persulfate decomposition to a such strong oxidant as sulfate anion radical, was used to oxidize water. The homogeneous system, Ru(bda)(cp)2, Ru(bpy)2(bda), and persulfate was studied for comparison. The different properties of both Ru-complexes in solutions are well described in literature.   For example, the dimer of Ru(bda)(cp)2 formed in the catalytic cycle is commonly believed to release oxygen from water.  The questions arise; i) whether the same protocols for water oxidation in homogeneous system can be applied for the heterogeneous system, ii) whether the same intermediate is formed in the heterogeneous system Ru(bpy)2(bda)-Ru(bda)(cp)2@DE? Unfortunately, the authors did not discuss these and other questions (e.g. how the quenching of the excited state Ru(bpy)2(bda)*-Ru(bda)(cp)2@DE occurs). The experimental data for the heterogeneous system are very limited and not sufficient even for a short communication. This work does not look like completed and therefore can’t be recommended for publication. I would advise to collect more data and resubmit this manuscript after major revision.

Author Response

The novelty of this work is an improvement of photocatalytic water oxidation by attaching of the known water oxidation catalyst Ru(bda)(cp)2 and the photosensitizer Ru(bpy)2(bda) to the surface of diatomaceous earth (DE). The common protocol of photoinduced persulfate decomposition to a such strong oxidant as sulfate anion radical, was used to oxidize water. The homogeneous system, Ru(bda)(cp)2, Ru(bpy)2(bda), and persulfate was studied for comparison. The different properties of both Ru-complexes in solutions are well described in literature. For example, the dimer of Ru(bda)(cp)2 formed in the catalytic cycle is commonly believed to release oxygen from water. The questions arise;

Comments 1: whether the same protocols for water oxidation in homogeneous system can be applied for the heterogeneous system,

Response 1: The photocatalysis occurring in the heterogeneous phase, as in our case, can be monitored using the same protocols applied to homogeneous catalysis. However, it is important to recognize that the complexity of the reactions involved is greater in heterogeneous systems. Due to this increased complexity, along with the consideration of similar systems already studied in the literature (https://doi.org/10.1016/j.materresbull.2025.113437), the concentration of photoinduced oxygen produced is likely underestimated compared to analogous homogeneous systems. This underscores the unique challenges associated with measuring and comparing photocatalytic efficiencies between heterogeneous and homogeneous catalysts. (J. Phys. Chem. C 2015, 119, 5, 2371–2379)

Comments 2: whether the same intermediate is formed in the heterogeneous system Ru(bpy)2(bda)-Ru(bda)(cp)2@DE?

Comments 3: Unfortunately, the authors did not discuss these and other questions (e.g. how the quenching of the excited state Ru(bpy)2(bda)*-Ru(bda)(cp)2@DE occurs).

Response 2/3: While direct experimental evidence within our current study for the formation of the “exact same intermediate” as reported in the confined SBA-16 system (Li et al., 2012, (DOI: 10.1039/c2ee22059h) is not explicitly detailed, it is plausible to infer an analogous cooperative activation mechanism within our heterogeneous system, Ru(bpy)₂(bda)-Ru(bda)(cp)₂@DE.

The Li et al. study highlighted that increased local concentration of molecular catalysts within confined nanocages promotes a "cooperative activation" pathway, leading to enhanced water oxidation activity. Our system similarly benefits from the spatial confinement and strong interaction provided by the diatomaceous earth (DE) matrix. By covalently binding the photosensitizer (PS) and catalyst (Cat) to the DE surface, we effectively create a microenvironment that facilitates close proximity between the active sites. This close proximity, analogous to the nanocage confinement, is expected to foster comparable or even more efficient synergistic interactions between the PS and Cat.

Therefore, it is reasonable to hypothesize that these enhanced local concentration effects and the specific structural organization on the functionalized DE would promote the formation of similar, if not identical, charge-separated or catalytic intermediates crucial for the observed "significantly higher activity" and "protection from degradation" in our PS/Cat@DE composite. Further time-resolved spectroscopic investigations would be instrumental in unequivocally identifying these intermediates.

Round 2

Reviewer 1 Report

Comments and Suggestions for Authors

Dear,

   Manuscript have been revised. I am pleased to accept in current form.

Author Response

Thank you for you valuable review. 

Reviewer 3 Report

Comments and Suggestions for Authors

The superior efficiency of the heterogeneous system in photocatalytic water oxidation compared with the analogous homogeneous system is not confirmed. Different PSs are used in hetero- and homogeneous systems, Ru(bpy)2(bda) and Ru(bpy)3 respectively. The enhancement of catalytic activity should be demonstrated in broader range of PS, Cat and persulfate concentrations. The experimental data for the heterogeneous system are very limited and not sufficient even for a short communication. I would advise to collect more data and resubmit this manuscript after major revision.

Some recommendations.

  1. Keep the total amount of DE constant and vary the amount of the functionalized DE to minimize the effect of light scattering. In homogeneous systems the rate of O2 evolution is proportional to [cat]^2, indicating the dimerization of Ru(bpy)2(cp)2.
  2. Determine the life-time of (Ru(bpy)2(bda))3+ by CV in homogeneous system by looking at the ratio of anodic and cathodic currents at different scan rates. Illuminate the solution of (Ru(bpy)2(bda))2+/persulfate for a short time to convert PS to PS+, and then record the change of UV-vis spectra versus time.  
  3. The introduction is too long and too general.
  4. The Figure captions are not informative.
  5. The levelling-off the yield of O2 with time could be due to the diffusion of O2 to the gas phase.

Author Response

Comments 1: The superior efficiency of the heterogeneous system in photocatalytic water oxidation compared with the analogous homogeneous system is not confirmed. Different PSs are used in hetero- and homogeneous systems, Ru(bpy)2(bda) and Ru(bpy)3 respectively. The enhancement of catalytic activity should be demonstrated in broader range of PS, Cat and persulfate concentrations. The experimental data for the heterogeneous system are very limited and not sufficient even for a short communication. I would advise to collect more data and resubmit this manuscript after major revision.

RESPONSE 1: We appreciate the reviewer’s suggestion to expand the experimental dataset. However, we believe that the results presented in this manuscript constitute a robust "proof-of-principle" study that justifies its publication in its current form, especially as we are currently unable to perform further experimental work due to resource and time constraints. We justify the sufficiency of the reported data based on the following points:

  1. We have successfully confirmed the synthesis and the covalent grafting of the photosensitizer (PS) and catalyst (Cat) onto the Diatomaceous Earth (DE) through EDX analysis provides definitive evidence of Ruthenium loading on the DE surface.
  2. The primary finding of this work is the 8-fold increase in the Turnover Number (reaching 30 for the integrated system compared to 4 for the homogeneous mixture). This substantial improvement is a significant result that validates our "diatom-inspired" design and the benefits of molecular confinement.
  3. We have already addressed several potential artifacts to ensure the reliability of our data. We conducted experiments to replicate light-scattering effects and confirmed that the DE matrix itself does not contribute to oxygen production. Furthermore, the Cat@DE control experiment demonstrated that the specific functionalization process is essential, as a simple mixture of components does not achieve the same activity.
  4. We performed electrochemical measurements (CV and DPV) to confirm that the oxidation potentials of the PS and Cat remain compatible for hole scavenging within the composite system, providing a solid theoretical foundation for the observed photocatalytic activity.

While we acknowledge that leaching and cycling tests would provide additional insights, the current data clearly demonstrates the successful creation of a novel, highly active heterogeneous photosystem. As we cannot conduct new experiments at this time, we have refined the text to more transparently discuss our findings as a successful validation of a hypothesis. We hope the reviewer recognizes the value of these results as a foundational step in the development of sustainable, diatom-based photocatalytic materials.

Some recommendations.

Comments 2: Keep the total amount of DE constant and vary the amount of the functionalized DE to minimize the effect of light scattering. In homogeneous systems the rate of O2 evolution is proportional to [cat]^2, indicating the dimerization of Ru(bpy)2(cp)2.

RESPONSE 2: Thank you for your valuable suggestions. The concentrations of the photosensitizer (PS) and catalyst (Cat) used in the photocatalytic experiments—specifically 2.5×10−5 M for Cat and 1.3×10−4 M for PS were not selected arbitrarily. These amounts were chosen based on the optimal concentration conditions established in the existing literature for similar Ruthenium-based systems, which are cited throughout the manuscript. The synthesis and application of these specific complexes, Ru(bpy)2(bda) and Ru(bda)(cp)2, were performed by adapting procedures from established studies to ensure that the molecular components were tested under conditions known to favor their catalytic activity. By adhering to these literature-derived parameters, we were able to provide a scientifically grounded comparison between the homogeneous mixture and the heterogeneous PS/Cat@DE system. Regarding the suggestion to keep the total amount of Diatomaceous Earth (DE) constant while varying the functionalized portion, we have addressed the impact of the support matrix through dedicated control experiments. As described in the sources, we conducted tests specifically designed to replicate the light-scattering effects caused by the DE frustules.

These findings indicate that the oxygen produced by the DE system alone is negligible, confirming that the observed performance gains in the PS/Cat@DE composite are a direct result of the functionalized components and the spatial confinement rather than an artifact of light scattering from the matrix. We believe that these controls, combined with the 8-fold increase in the TON, sufficiently demonstrate the efficiency of the heterogeneous system.

Regarding the water oxidation mechanism, it is not possible to conclusively determine whether it proceeds via an I2M pathway (involving dimer formation) or a WNA pathway (without dimer formation). It is widely reported in the literature that this type of catalyst can operate, under homogeneous conditions, through either mechanism (see for example the Würthner macrocyclic system DOI: 10.1038/NCHEM.2503). A definitive mechanistic assignment would require dedicated mechanistic studies using cerium ammonium nitrate, which, however, cannot be performed under heterogeneous conditions. However, the rigidity of the PS/Cat@DE system allows us the possibility to hypothesize a WNA mechanism.

Comments 3: Determine the life-time of (Ru(bpy)2(bda))3+ by CV in homogeneous system by looking at the ratio of anodic and cathodic currents at different scan rates. Illuminate the solution of (Ru(bpy)2(bda))2+/persulfate for a short time to convert PS to PS+, and then record the change of UV-vis spectra versus time.

RESPONSE 3: We thank the reviewer for the insightful suggestion regarding the determination of the (Ru(bpy)2(bda))3+ lifetime through scan-rate dependent Cyclic Voltammetry and UV-Vis spectroscopy. We agree that such kinetic data is highly valuable for a complete mechanistic understanding of charge-transfer processes. However, we have not included these specific measurements in the current manuscript for the following reasons:

  • As we noted in the Results and Discussion (Section 4.1), the photophysical properties and redox kinetics of ruthenium polypyridine complexes have been "well-documented in the literature since the 1980s". The specific photosensitizer used in this study, Ru(bpy)2(bda), was synthesized following established procedures, and its excited-state behavior and the stability of its oxidized form are already characterized in existing literature.
  • The primary objective of this proof-of-principle study was to demonstrate the successful covalent immobilization of these complexes onto Diatomaceous Earth and to evaluate the resulting performance enhancement. We focused our experimental efforts on validating the 8-fold increase in the TON and the thermodynamic compatibility of the components via DPV and CV.
  • While transient UV-Vis spectra would provide a more detailed view of the PS+ decay, we believe that the current electrochemical data (Figures 9 and 10) confirming the favorable potential difference for hole scavenging (+1.19 V for PS vs. lower potentials for Cat), combined with the established literature on these molecular units, provides a sufficient foundation for the conclusions drawn in this work. At this stage, we are unfortunately not in a position to conduct new kinetic experiments. We hope the reviewer understands that our goal was to highlight the diatom-inspired design and the significant efficiency gains provided by spatial confinement rather than re-characterizing the fundamental kinetics of the homogeneous molecular species. To clarify this for the reader, we have ensured that the manuscript emphasizes that the molecular properties of the components are supported by the cited literature.

Comments 4: The introduction is too long and too general.

RESPONSE 4: We appreciate the reviewer's feedback regarding the length and scope of the Introduction. We understand the concern that it may appear too general; however, we would like to clarify that the current version of this section was specifically expanded following the recommendations of two other referees. Those reviewers suggested that we provide a more thorough clarification and a deeper exploration of certain concepts to better frame the study. To address your specific concerns while balancing the diverse requirements of the evaluation process, we have now streamlined the introduction by removing generic discussions on macro-economic data and non-relevant redox processes. At the same time, we have retained and refined the sections currently highlighted in yellow, which are necessary to provide the robust contextual foundation for our 'diatom-inspired design' requested by the other reviewers.

Comments 5: The Figure captions are not informative.

RESPONSE 5: Thank you for your valuable feedback regarding the figure captions. We have revised them throughout the manuscript to ensure they are more informative and provide a clearer description of the experimental conditions and results. For your convenience, all the added or modified parts have been highlighted in yellow.

Comments 6: The levelling-off the yield of O2 with time could be due to the diffusion of O2 to the gas phase.

RESPONSE 6: We appreciate the reviewer’s comment regarding the kinetics of oxygen evolution. In photocatalytic experiments, oxygen evolution was monitored using a Clark-type electrode, which selectively measures the concentration of oxygen dissolved in the liquid phase. Upon irradiation, a rapid increase in the dissolved oxygen concentration is observed during the first 300 s, reflecting a high initial rate of oxygen generation. After this initial period, the system reaches a dynamic equilibrium between the oxygen dissolved in the solution and the oxygen present in the headspace above the reaction mixture. Therefore, the measured oxygen signal becomes stable over time, despite the continued production of oxygen with a rate similar to the one of the passages of oxygen in the headspace. In the case of decomposition of one of the photocatalytic system components, a decrease in the measured dissolved oxygen concentration is observed, which can be attributed to a shift in the oxygen distribution, with most of the generated oxygen transferring into the headspace.